# Multimodal Chain-of-Thought Reasoning in Language Models

## Abstract

Large language models (LLMs) have shown impressive performance on complex reasoning by leveraging chain-of-thought (CoT) prompting to generate intermediate reasoning chains as the rationale to infer the answer. However, existing CoT studies have primarily focused on the language modality. We propose Multimodal-CoT that incorporates language (text) and vision (images) modalities into a two-stage framework that separates rationale generation and answer inference. In this way, answer inference can leverage better generated rationales that are based on multimodal information. Experimental results on ScienceQA and A-OKVQA benchmark datasets show the effectiveness of our proposed approach. With Multimodal-CoT, our model under 1 billion parameters achieves new state-of-the-art performance on the ScienceQA benchmark. Our analysis indicates that Multimodal-CoT offers the advantages of mitigating hallucination. Code is publicly available at Anonymous.

## 1 Introduction

Imagine reading a textbook with no figures or tables. Our ability to knowledge acquisition is greatly strengthened by jointly modeling diverse data modalities, such as vision, language, and audio. Recently, large language models (LLMs) (Brown et al., 2020; Thoppilan et al., 2022; Rae et al., 2021; Chowdhery et al., 2022) have shown impressive performance in complex reasoning by generating intermediate reasoning steps before inferring the answer. The intriguing technique is called chain-of-thought (CoT) reasoning (Wei et al., 2022b; Kojima et al., 2022; Zhang et al., 2023c).

However, existing studies related to CoT reasoning are largely isolated in the language modality (Wang et al., 2022c; Zhou et al., 2022; Lu et al., 2022b; Fu et al., 2022), with little consideration of multimodal scenarios. To elicit CoT reasoning in multimodality, we advocate a Multimodal-CoT paradigm. Given the inputs in different modalities, Multimodal-CoT decomposes multistep problems into intermediate reasoning steps (rationale) and then infers the answer. Since vision and language are the most popular modalities, we focus on those two modalities in this work. An example is shown in Figure 1.

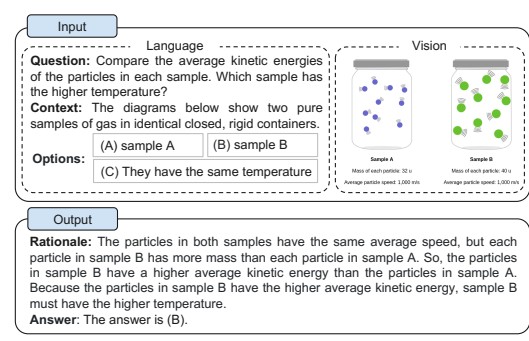

Figure 1: Example of the multimodal CoT task.

In general, there are two ways to elicit Multimodal-CoT reasoning as follows: (i) prompting LLMs and (ii) fine-tuning small models.[1]

The most immediate way to perform Multimodal-CoT is to transform the input of different modalities into a unified modality and prompt LLMs to perform CoT (Zhang et al., 2023a; Lu et al., 2023; Liu et al., 2023; Alayrac et al., 2022; Hao et al., 2022; Yasunaga et al., 2022). For example, it is possible to generate a caption for an image by a captioning model and then concatenate the caption with the original language input to be fed into LLMs (Lu et al., 2022a). However, there is severe information loss in the captioning process; thus, using image captions (as opposed to vision features) may suffer

---

[1]We refer to small models as models with less than 1 billion parameters (hereinafter dubbed as 1B-models).

from a lack of mutual synergy in the representation space of different modalities. In addition, LLMs either have paywalls or resource-consuming to deploy locally.

To facilitate the interaction between modalities, another potential solution is to fine-tune smaller language models (LMs) by fusing multimodal features (Zhang et al., 2023b). As this approach allows the flexibility of adjusting model architectures to incorporate multimodal features, we study fine-tuning models in this work instead of prompting LLMs. The key challenge is that language models under 100 billion parameters tend to generate hallucinated rationales that mislead the answer inference (Ho et al., 2022; Magister et al., 2022; Ji et al., 2022).

To mitigate the challenge of hallucination, we propose Multimodal-CoT that incorporates language (text) and vision (images) modalities into a two-stage framework that separates rationale generation and answer inference.[2] In this way, answer inference can leverage better generated rationales that are based on multimodal information. Our experiments are conducted on the ScienceQA (Lu et al., 2022a) and A-OKVQA (Schwenk et al., 2022) datasets, which are the latest multimodal reasoning benchmarks with annotated reasoning chains.

Our method achieves new state-of-the-art performance on the ScienceQA benchmark. We find that Multimodal-CoT is beneficial in mitigating hallucination and boosting convergence. Our contributions are summarized as follows:

(i) To the best of our knowledge, this work is the first to study CoT reasoning in different modalities in scientific peer-reviewed literature.

(ii) We propose a two-stage framework by fine-tuning language models to fuse vision and language representations to perform Multimodal-CoT. The model is able to generate informative rationales to facilitate inferring final answers.

(iii) Our method achieves new state-of-the-art performance on the ScienceQA benchmark. Our work elicits the analysis of why the naive way of employing CoT fails in the context and how incorporating vision features alleviates the problem. The approach has been shown to be generally effective across tasks and backbone models.

## 2 BACKGROUND

This section reviews studies eliciting CoT reasoning by prompting and fine-tuning language models.

### 2.1 CoT REASONING WITH LLMS

Recently, CoT has been widely used to elicit the multi-step reasoning abilities of LLMs (Wei et al., 2022b). Concretely, CoT techniques encourage the LLM to generate intermediate reasoning chains for solving a problem. Studies have shown that LLMs can perform CoT reasoning with two major paradigms of techniques: Zero-Shot-CoT (Kojima et al., 2022) and Few-Shot-CoT (Wei et al., 2022b; Zhang et al., 2023c). For Zero-Shot-CoT, Kojima et al. (2022) showed that LLMs are decent zero-shot reasoners by adding a prompt like "Let's think step by step" after the test question to invoke CoT reasoning. For Few-Shot-CoT, a few step-by-step reasoning demonstrations are used as conditions for inference. Each demonstration has a question and a reasoning chain that leads to the final answer. The demonstrations are commonly obtained by hand-crafting or automatic generation. These two techniques, hand-crafting and automatic generation are thus referred to as Manual-CoT (Wei et al., 2022b) and Auto-CoT (Zhang et al., 2023c).

With effective demonstrations, Few-Shot-CoT often achieves stronger performance than Zero-Shot-CoT and has attracted more research interest. Therefore, most recent studies focused on how to improve Few-Shot-CoT. Those studies are categorized into two major research lines: (i) optimizing the demonstrations; (ii) optimizing the reasoning chains. Table 1 compares typical CoT techniques.

**Optimizing Demonstrations** The performance of Few-Shot-CoT relies on the quality of demonstrations. As reported in Wei et al. (2022b), using demonstrations written by different annotators results in dramatic accuracy disparity in reasoning tasks. Beyond hand-crafting the demonstrations,

---

[2]This work focuses on the language and vision modalities.

Table 1: Representative CoT techniques (FT: fine-tuning; KD: knowledge distillation). Segment 1: in-context learning techniques; Segment 2: fine-tuning techniques. To the best of our knowledge, our work is the first to study CoT reasoning in different modalities in scientific peer-reviewed literature. Besides, we focus on 1B-models, without relying on the outputs of LLMs.

| Models | Mutimodal | Model / Engine | Training | CoT Role | CoT Source |
|---|---|---|---|---|---|
| Zero-Shot-CoT (Kojima et al., 2022) | ✗ | GPT-3.5 (175B) | ICL | Reasoning | Template |
| Few-Shot-CoT (Wei et al., 2022b) | ✗ | PaLM (540B) | ICL | Reasoning | Hand-crafted |
| Self-Consistency-CoT (Wang et al., 2022b) | ✗ | Codex (175B) | ICL | Reasoning | Hand-crafted |
| Least-to-Most Prompting (Zhou et al., 2022) | ✗ | Codex (175B) | ICL | Reasoning | Hand-crafted |
| Retrieval-CoT (Zhang et al., 2023c) | ✗ | GPT-3.5 (175B) | ICL | Reasoning | Auto-generated |
| PromptPG-CoT (Lu et al., 2022b) | ✗ | GPT-3.5 (175B) | ICL | Reasoning | Hand-crafted |
| Auto-CoT (Zhang et al., 2023c) | ✗ | Codex (175B) | ICL | Reasoning | Auto-generated |
| Complexity-CoT (Fu et al., 2022) | ✗ | GPT-3.5 (175B) | ICL | Reasoning | Hand-crafted |
| Few-Shot-PoT (Chen et al., 2022) | ✗ | GPT-3.5 (175B) | ICL | Reasoning | Hand-crafted |
| UnifiedQA (Lu et al., 2022a) | ✗ | T5 (770M) | FT | Explanation | Crawled |
| Fine-Tuned T5 XXL (Magister et al., 2022) | ✗ | T5 (11B) | KD | Reasoning | LLM-generated |
| Fine-Tune-CoT (Ho et al., 2022) | ✗ | GPT-3 (6.7B) | KD | Reasoning | LLM-generated |
| Multimodal-CoT (our work) | ✓ | T5 (770M) | FT | Reasoning | Crawled |

recent studies have investigated ways to optimize the demonstration selection process. Notably, Rubin et al. (2022) retrieved the semantically similar demonstrations with the test instance. However, this approach shows a degraded performance when there are mistakes in the reasoning chains (Zhang et al., 2023c). To address the limitation, Zhang et al. (2023c) found that the key is the diversity of demonstration questions and proposed Auto-CoT: (i) partition questions of a given dataset into a few clusters; (ii) sample a representative question from each cluster and generate its reasoning chain using Zero-Shot-CoT with simple heuristics. In addition, reinforcement learning (RL) and complexity-based selection strategies were proposed to obtain effective demonstrations. Fu et al. (2022) chose examples with complex reasoning chains (i.e., with more reasoning steps) as the demonstrations. Lu et al. (2022b) trained an agent to find optimal in-context examples from a candidate pool and maximize the prediction rewards on given training examples when interacting with GPT-3.5.

**Optimizing Reasoning Chains** A notable way to optimize reasoning chains is problem decomposition. Zhou et al. (2022) proposed least-to-most prompting to decompose complex problems into sub-problems and then solve these sub-problems sequentially. As a result, solving a given sub-problem is facilitated by the answers to previously solved sub-problems. Similarly, Khot et al. (2022) used diverse decomposition structures and designed different prompts to answer each sub-question. In addition to prompting the reasoning chains as natural language texts, Chen et al. (2022) proposed program-of-thoughts (PoT), which modeled the reasoning process as a program and prompted LLMs to derive the answer by executing the generated programs. Another trend is to vote over multiple reasoning paths for a test question. Wang et al. (2022b) introduced a self-consistency decoding strategy to sample multiple outputs of LLMs and then took a majority over the final answers. Wang et al. (2022c) and Li et al. (2022c) introduced randomness in the input space to produce more diverse outputs for voting.

## 2.2 ELICITING COT REASONING BY FINE-TUNING MODELS

A recent interest is eliciting CoT reasoning by fine-tuning language models. Lu et al. (2022a) fine-tuned the encoder-decoder T5 model on a large-scale dataset with CoT annotations. However, a dramatic performance decline is observed when using CoT to infer the answer, i.e., generating the reasoning chain before the answer (reasoning). Instead, CoT is only used as an explanation after the answer. Magister et al. (2022) and Ho et al. (2022) employed knowledge distillation by fine-tuning a student model on the chain-of-thought outputs generated by a larger teacher model. Wang et al. (2022a) proposed an iterative context-aware prompting approach to dynamically synthesize prompts conditioned on the current step's contexts.

There is a key challenge in training 1B-models to be CoT reasoners. As observed by Wei et al. (2022b), models under 100 billion parameters tend to produce illogical CoT that leads to wrong answers. In other words, it might be harder for 1B-models to generate effective CoT than directly generating the answer. It becomes even more challenging in a multimodal setting where answering

the question also requires understanding the multimodal inputs. In the following part, we will explore the challenge of Multimodal-CoT and investigate how to perform effective multi-step reasoning.

# 3 CHALLENGE OF MULTIMODAL-COT

Existing studies have suggested that the CoT reasoning ability may emerge in language models at a certain scale, e.g., over 100 billion parameters (Wei et al., 2022a). However, it remains an unresolved challenge to elicit such reasoning abilities in 1B-models, let alone in the multimodal scenario. This work focuses on 1B-models as they can be fine-tuned and deployed with consumer-grade GPUs (e.g., 32G memory). In this section, we will investigate why 1B-models fail at CoT reasoning and study how to design an effective approach to overcome the challenge.

## 3.1 TOWARDS THE ROLE OF COT

To begin with, we fine-tune a text-only baseline for CoT reasoning on the ScienceQA benchmark (Lu et al., 2022a). We adopt FLAN-Alpaca$_{Base}$ as the backbone language model.[3] Our task is modeled as a text generation problem, where the model takes the textual information as the input and generates the output sequence that consists of the rationale and the answer. As an example shown in Figure 1, the model takes the concatenation of tokens of the question text (Q), the context text (C), and multiple options (M) as the input. To study the effect of CoT, we compare the performance with three variants: (i) `No-CoT` which predicts the answer directly (QCM→A); (ii) `Reasoning` where answer inference is conditioned to the rationale (QCM→RA); (iii) `Explanation` where the rationale is used for explaining the answer inference (QCM→AR).

Surprisingly, we observe a ↓12.31% accuracy decrease (81.63%→69.32%) if the model predicts rationales before answers (QCM→RA). The results imply that the rationales might not necessarily contribute to predicting the right answer. According to Lu et al. (2022a), the plausible reason might be that the model exceeds the maximum token limits before obtaining the required answer or stops generating the prediction early. However, we find that the maximum length of the generated outputs (RA) is always less than 400 tokens, which is below the length limit of language models (i.e., 512 in T5 models). Therefore, it deserves a more in-depth investigation into why the rationales harm answer inference.

Table 2: Effects of CoT in the one-stage setting.

| Method | Format | Accuracy |
|---|---|---|
| No-CoT | QCM→A | 81.63 |
| Reasoning | QCM→RA | 69.32 |
| Explanation | QCM→AR | 69.68 |

## 3.2 MISLEADING BY HALLUCINATED RATIONALES

To dive into how the rationales affect the answer prediction, we separate the CoT problem into two stages, *rationale generation* and *answer inference*.[4] We report the RougeL score and accuracy for the rationale generation and answer inference, respectively. Table 3 shows the results based on the two-stage framework. Although the two-stage baseline model achieves a 90.73 RougeL score of the rationale generation, the answer inference accuracy is only 78.57%. Compared with the QCM→A variant (81.63%) in Table 2, the result shows that the generated rationale in the two-stage framework does not improve answer accuracy.

Table 3: Two-stage setting of (i) rationale generation (RougeL) and (ii) answer inference (Accuracy).

| Method | (i) QCM→ R | (ii) QCMR→ A |
|---|---|---|
| Two-Stage Framework | 90.73 | 78.57 |
| w/ Captions | 90.88 | 79.37 |
| w/ Vision Features | 93.46 | 85.31 |

Then, we randomly sample 50 error cases and find that the model tends to generate hallucinated rationales that mislead the answer inference. As an example shown in Figure 2, the model (left part) hallucinates that, "*The south pole of one magnet is closest to the south pole of the other magnet*",

---

[3] `https://github.com/declare-lab/flan-alpaca`. It is a 200M T5 model (Raffel et al., 2020) fine-tuned on Stanford Alpaca data (Taori et al., 2023). Implementation details are presented in Appendix B.1.

[4] The details will be presented in Section 4.

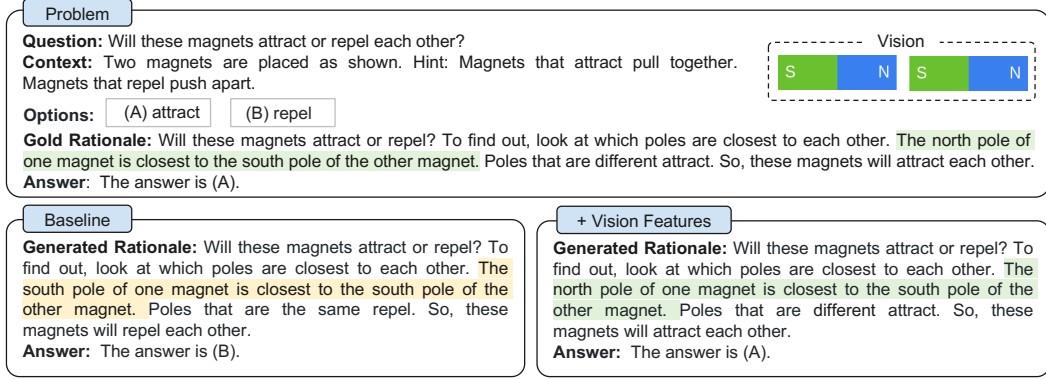

Figure 2: Example of the two-stage framework without vision features (baseline) and with vision features (ours) for generating rationales and predicting answers. The upper part presents the problem details with a gold rationale, and the lower part shows the outputs of the baseline and our method incorporated with vision features. We observe that the baseline fails to predict the right answer due to the misleading by hallucinated rationales. More examples are shown in Appendix A.1.

due to the lack of reference to the vision content. We find that such mistakes occur at a ratio of 56% among the error cases (Figure 3(a)).

## 3.3 MULTIMODALITY CONTRIBUTES TO EFFECTIVE RATIONALES

We speculate that such a phenomenon of hallucination is due to a lack of necessary vision contexts for performing effective Multimodal-CoT. To inject vision information, a simple way is to transform the image into a caption (Lu et al., 2022a) and then append the caption in the input of both stages. However, as shown in Table 3, using captions only yields marginal performance gains (↑0.80%). Then, we explore an advanced technique by incorporating vision features into the language model. Concretely, we feed the image to the ViT model (Dosovitskiy et al., 2021b) to extract vision features. Then we fuse the vision features with the encoded language representations before feeding the decoder (more details will be presented in Section 4). Interestingly, with vision features, the RougeL score of the rationale generation has boosted to 93.46% (QCM→R), which correspondingly contributes to better answer accuracy of 85.31% (QCMR→A).

With those effective rationales, the phenomenon of hallucination is mitigated — 60.7% hallucination mistakes in Section 3.2 have been corrected (Figure 3(b)), as an example shown in Figure 2 (right part).[5] The analysis so far compellingly shows that vision features are indeed beneficial for generating effective rationales and contributing to accurate answer inference. As the two-stage method achieves better performance than one-stage methods, we choose the two-stage method in our Multimodal-CoT framework.

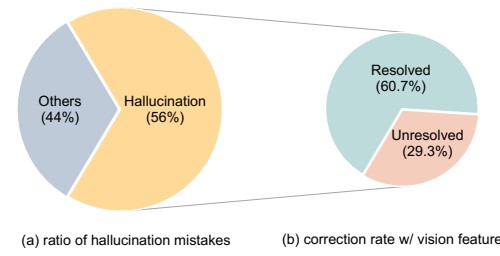

(a) ratio of hallucination mistakes   (b) correction rate w/ vision features

Figure 3: The ratio of (a) hallucination mistakes and (b) correction rate w/ vision features.

## 4 MULTIMODAL-COT

In light of the discussions in Section 3, we propose Multimodal-CoT to incorporate language (text) and vision (images) modalities into a two-stage framework. The key motivation is the anticipation that the answer inference can leverage better generated rationales that are based on multimodal information. In this section, we will overview the procedure of the framework and elaborate on the technical design of the model architecture.

---

[5]The left mistakes are mainly about map understanding, requiring extra commonsense signals (Section 6.3).

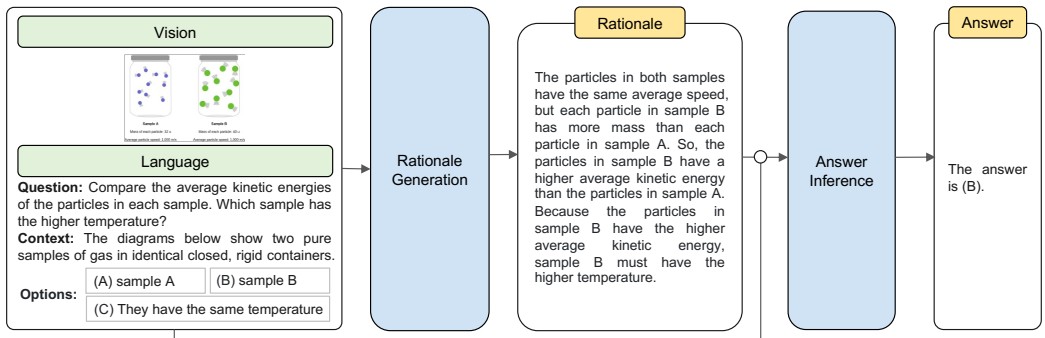

Figure 4: Overview of our Multimodal-CoT framework. Multimodal-CoT consists of two stages: (i) rationale generation and (ii) answer inference. Both stages share the same model structure but differ in the input and output. In the first stage, we feed the model with language and vision inputs to generate rationales. In the second stage, we append the original language input with the rationale generated from the first stage. Then, we feed the updated language input with the original vision input to the model to infer the answer.

## 4.1 FRAMEWORK OVERVIEW

Multimodal-CoT consists of two operation stages: (i) rationale generation and (ii) answer inference. Both stages share the same model structure but differ in the input $X$ and output $Y$. The overall procedure is illustrated in Figure 4. We will take vision-language as an example to show how Multimodal-CoT works.

In the rationale generation stage, we feed the model with $X = \{X^1_{\text{language}}, X_{\text{vision}}\}$ where $X^1_{\text{language}}$ represents the language input in the first stage and $X_{\text{vision}}$ represents the vision input, i.e., the image. For example, $X$ can be instantiated as a concatenation of question, context, and options of a multiple choice reasoning problem (Lu et al., 2022a) as shown in Figure 4. The goal is to learn a rationale generation model $R = F(X)$ where $R$ is the rationale.

In the answer inference stage, the rationale $R$ is appended to the original language input $X^1_{\text{language}}$ to construct the language input in the second stage, $X^2_{\text{language}} = X^1_{\text{language}} \circ R$ where $\circ$ denotes concatenation. Then, we feed the updated input $X' = \{X^2_{\text{language}}, X_{\text{vision}}\}$ to the answer inference model to infer the final answer $A = F(X')$.

In both stages, we train two models with the same architecture independently. They take the annotated elements (e.g., $X \to R$, $XR \to A$, respectively) from the training set for supervised learning. During inference, given $X$, the rationales for the test sets are generated using the model trained in the first stage; they are used in the second stage for answer inference.

## 4.2 MODEL ARCHITECTURE

Given language input $X_{\text{language}} \in \{X^1_{\text{language}}, X^2_{\text{language}}\}$ and vision input $X_{\text{vision}}$, we compute the probability of generating target text $Y$ (either the rationale or the answer in Figure 4) of length $N$ by

$$p(Y|X_{\text{language}}, X_{\text{vision}}) = \prod_{i=1}^{N} p_\theta \left(Y_i \mid X_{\text{language}}, X_{\text{vision}}, Y_{<i}\right), \tag{1}$$

where $p_\theta \left(Y_i \mid X_{\text{language}}, X_{\text{vision}}, Y_{<i}\right)$ is implemented with a Transformer-based network (Vaswani et al., 2017). The network has three major procedures: encoding, interaction, and decoding. Specifically, we feed the language text into a Transformer encoder to obtain a textual representation, which is interacted and fused with the vision representation before being fed into the Transformer decoder.

**Encoding**  The model $F(X)$ takes both the language and vision inputs and obtains the text representation $H_{\text{language}}$ and the image feature $H_{\text{vision}}$ by the following functions:

$$H_{\text{language}} = \text{LanguageEncoder}(X_{\text{language}}), \tag{2}$$

$$H_{\text{vision}} = W_h \cdot \text{VisionExtractor}(X_{\text{vision}}), \tag{3}$$

where LanguageEncoder(·) is implemented as a Transformer model. We use the hidden states of the last layer in the Transformer encoder as the language representation $H_{\text{language}} \in \mathbb{R}^{n \times d}$ where $n$ denotes the length of the language input, and $d$ is the hidden dimension. Meanwhile, VisionExtractor(·) is used to vectorize the input image into vision features. Inspired by the recent success of Vision Transformers (Dosovitskiy et al., 2021a), we fetch the patch-level features by frozen vision extraction models, such as ViT (Dosovitskiy et al., 2021b). After obtaining the patch-level vision features, we apply a learnable projection matrix $W_h$ to convert the shape of VisionExtractor($X_{\text{vision}}$) into that of $H_{\text{language}}$; thus we have $H_{\text{vision}} \in \mathbb{R}^{m \times d}$ where $m$ is the number of patches.

Note that our approach is general to both scenarios with or without image context. For the questions without associated images, we use all-zero vectors as the "blank features" with the same shape as the normal image features to tell the model to ignore them.

**Interaction** After obtaining language and vision representations, we use a single-head attention network to correlate text tokens with image patches, where the query ($Q$), key ($K$) and value ($V$) are $H_{\text{language}}$, $H_{\text{vision}}$ and $H_{\text{vision}}$, respectively. The attention output $H_{\text{vision}}^{\text{attn}} \in \mathbb{R}^{n \times d}$ is defined as:

$$H_{\text{vision}}^{\text{attn}} = \text{Softmax}(\frac{QK^{\top}}{\sqrt{d_k}})V, \tag{4}$$

where $d_k$ is the same as the dimension of $H_{\text{language}}$ because a single head is used.

Then, we apply the gated fusion mechanism (Zhang et al., 2020; Wu et al., 2021; Li et al., 2022a) to fuse $H_{\text{language}}$ and $H_{\text{vision}}$. The fused output $H_{\text{fuse}} \in \mathbb{R}^{n \times d}$ is obtained by:

$$\lambda = \text{Sigmoid}(W_l H_{\text{language}} + W_v H_{\text{vision}}^{\text{attn}}), \tag{5}$$

$$H_{\text{fuse}} = (1 - \lambda) \cdot H_{\text{language}} + \lambda \cdot H_{\text{vision}}^{\text{attn}}, \tag{6}$$

where $W_l$ and $W_v$ are learnable parameters.

**Decoding** Finally, the fused output $H_{\text{fuse}}$ is fed into the Transformer decoder to predict the target $Y$.

## 5 EXPERIMENTS

This section will present the benchmark dataset, the implementation of our technique, and the baselines for comparisons. Then, we will report our main results and findings.

### 5.1 DATASET

Our method is evaluated on the ScienceQA (Lu et al., 2022a) and A-OKVQA (Schwenk et al., 2022) benchmark datasets. ScienceQA is a large-scale multimodal science question dataset with annotated lectures and explanations. It contains $21k$ multimodal multiple choice questions with rich domain diversity across 3 subjects, 26 topics, 127 categories, and 379 skills. There are $12k$, $4k$, and $4k$ questions in the training, validation, and test splits, respectively. A-OKVQA is a knowledge-based visual question answering benchmark, which has $25k$ questions requiring a broad base of commonsense and world knowledge to answer. It has $17k/1k/6k$ questions for train/val/test. To keep consistency with ScienceQA, we use the multiple-choice setting.

### 5.2 IMPLEMENTATION

The following part presents the experimental settings of Multimodal-CoT and the baseline methods.

**Experimental Settings** We adopt the T5 encoder-decoder architecture (Raffel et al., 2020) under `Base` (200M) and `large` (700M) settings in our framework. We apply FLAN-Alpaca to initialize our model weights.[6] We will show that Multimodal-CoT is generally effective with other backbone LMs, such as UnifiedQA (Khashabi et al., 2020) and FLAN-T5 (Chung et al., 2022) (Section 6.1). The vision features are obtained by the frozen ViT-large encoder (Dosovitskiy et al., 2021b). We fine-tune the models up to 20 epochs, with a learning rate of 5e-5. The maximum input sequence length is 512. The batch size is 8. Our experiments are run on 8 NVIDIA Tesla V100 32G GPUs.

---

[6]https://github.com/declare-lab/flan-alpaca.

Table 4: Main results (%). Size = backbone model size from the ScienceQA leaderboard ("-" means unavailable or unknown). Question classes: NAT = natural science, SOC = social science, LAN = language science, TXT = text context, IMG = image context, NO = no context, G1-6 = grades 1-6, G7-12 = grades 7-12. Segment 1: Human performance; Segment 2: VQA baselines; Segment 3: LM baselines, i.e., UnifiedQA and few-shot learning LLMs; Segment 4: Fine-tuned large vision-language models; Segment 5: Our Multimodal-CoT results. Prior published best results are marked with an underline. Our best average result is in **bold** face. † denotes concurrent studies after this work.

| Model | Size | NAT | SOC | LAN | TXT | IMG | NO | G1-6 | G7-12 | Avg |
|---|---|---|---|---|---|---|---|---|---|---|
| Human | - | 90.23 | 84.97 | 87.48 | 89.60 | 87.50 | 88.10 | 91.59 | 82.42 | 88.40 |
| MCAN (Yu et al., 2019) | 95M | 56.08 | 46.23 | 58.09 | 59.43 | 51.17 | 55.40 | 51.65 | 59.72 | 54.54 |
| Top-Down (Anderson et al., 2018) | 70M | 59.50 | 54.33 | 61.82 | 62.90 | 54.88 | 59.79 | 57.27 | 62.16 | 59.02 |
| BAN (Kim et al., 2018) | 112M | 60.88 | 46.57 | 66.64 | 62.61 | 52.60 | 65.51 | 56.83 | 63.94 | 59.37 |
| DFAF (Gao et al., 2019) | 74M | 64.03 | 48.82 | 63.55 | 65.88 | 54.49 | 64.11 | 57.12 | 67.17 | 60.72 |
| ViLT (Kim et al., 2021) | 113M | 60.48 | 63.89 | 60.27 | 63.20 | 61.38 | 57.00 | 60.72 | 61.90 | 61.14 |
| Patch-TRM (Lu et al., 2021) | 90M | 65.19 | 46.79 | 65.55 | 66.96 | 55.28 | 64.95 | 58.04 | 67.50 | 61.42 |
| VisualBERT (Li et al., 2019) | 111M | 59.33 | 69.18 | 61.18 | 62.71 | 62.17 | 58.54 | 62.96 | 59.92 | 61.87 |
| UnifiedQA (Lu et al., 2022a) | 223M | 71.00 | 76.04 | 78.91 | 66.42 | 66.53 | 81.81 | 77.06 | 68.82 | 74.11 |
| GPT-3.5 (text-davinci-002) (Lu et al., 2022a) | 173B | 75.44 | 70.87 | 78.09 | 74.68 | 67.43 | 79.93 | 78.23 | 69.68 | 75.17 |
| GPT-3.5 (text-davinci-003) | 173B | 77.71 | 68.73 | 80.18 | 75.12 | 67.92 | 81.81 | 80.58 | 69.08 | 76.47 |
| ChatGPT (Lu et al., 2023) | - | 78.82 | 70.98 | 83.18 | 77.37 | 67.92 | 86.13 | 80.72 | 74.03 | 78.31 |
| GPT-4 (Lu et al., 2023) | - | 85.48 | 72.44 | 90.27 | 82.65 | 71.49 | 92.89 | 86.66 | 79.04 | 83.99 |
| Chameleon (ChatGPT) (Lu et al., 2023)† | - | 81.62 | 70.64 | 84.00 | 79.77 | 70.80 | 86.62 | 81.86 | 76.53 | 79.93 |
| Chameleon (GPT-4) (Lu et al., 2023)† | - | 89.83 | 74.13 | 89.82 | 88.27 | 77.64 | 92.13 | 88.03 | 83.72 | 86.54 |
| LLaMA-Adapter (Zhang et al., 2023a)† | 6B | 84.37 | 88.30 | 84.36 | 83.72 | 80.32 | 86.90 | 85.83 | 84.05 | 85.19 |
| LLaVA (Liu et al., 2023)† | 13B | 90.36 | 95.95 | 88.00 | 89.49 | 88.00 | 90.66 | 90.93 | 90.90 | 90.92 |
| InstructBLIP (Dai et al., 2023)† | 11B | - | - | - | - | 90.70 | - | - | - | |
| Mutimodal-CoT$_{Base}$ | 223M | 84.06 | 92.35 | 82.18 | 82.75 | 82.75 | 84.74 | 85.79 | 84.44 | 85.31 |
| Mutimodal-CoT$_{Large}$ | 738M | 91.03 | 93.70 | 86.64 | 90.13 | 88.25 | 89.48 | 91.12 | 89.26 | **90.45** |

**Baseline Models** Our baselines include (i) Visual question answering (VQA) models (Anderson et al., 2018; Kim et al., 2018; Yu et al., 2019; Gao et al., 2019; Kim et al., 2021; Lu et al., 2021; Li et al., 2019); (ii) LMs, including the Text-to-text UnifiedQA model (Khashabi et al., 2020) and few-shot learning LLMs (GPT-3.5, ChatGPT, GPT-4, and Chameleon (Lu et al., 2023)); (iii) Fine-tuned large vision-language model LLaMA-Adapter (Zhang et al., 2023a), LLaVA (Liu et al., 2023), and InstructBLIP (Dai et al., 2023). More details are presented in Appendix B.1.

## 5.3 MAIN RESULTS

Table 4 shows the main results in the ScienceQA benchmark. Mutimodal-CoT$_{Large}$ achieves substantial performance gains over the prior best model in publications (86.54%→90.45%). The efficacy of Multimodal-CoT is further supported by the results obtained from the A-OKVQA benchmark (Table 5). Our ablation study (Appendix C.1) reveals that both the integration of vision features and the two-stage framework design contribute to the overall performance. Furthermore, Multimodal-CoT demonstrates the ability to mitigate hallucination (Section 3.3) and improve convergence (Appendix C.2).

It is worth noting that Chameleon, LLaMA-Adapter, LLaVA, and InstructBLIP are concurrent works released several months after our work. We show that our method

Table 5: Results on the A-OKVQA dataset. Baseline results are from (Chen et al., 2023) and Schwenk et al. (2022).

| Model | Accuracy |
|---|---|
| BERT | 32.93 |
| GPT-3 (Curie) | 35.07 |
| IPVR (OPT-66B) | 48.6 |
| ViLBERT | 49.1 |
| LXMERT | 51.4 |
| Language-only Baseline | 47.86 |
| Multimodal-CoT$_{Base}$ | 50.57 |

is orthogonal to those latest multimodal models (e.g., InstructBLIP) and can be potentially used with them together to improve generality further, i.e., scaled to scenarios where human-annotated rationales are unavailable (Appendix C.3), thereby establishing the effectiveness across diverse tasks.

## 6 ANALYSIS

The following analysis will investigate whether Multimodal-CoT is generally effective with different backbone models and vision features. We will also conduct an error analysis to explore the limitations to inspire future studies. We use models under the `base` size for analysis unless otherwise stated.

## 6.1 Effectiveness Across Backbones

To test the generality of the benefits of our approach to other backbone models, we alter the underlying LMs to other variants in different types. As shown in Table 6, our approach is generally effective for the widely used backbone models.

Table 6: Using different backbone LMs. More detailed results are presented in Appendix C.5.

| Method | Accuracy |
|---|---|
| Prior Best (Lu et al., 2022a) | 75.17 |
| MM-CoT on UnifiedQA | 82.55 |
| MM-CoT on FLAN-T5 | 83.19 |
| MM-CoT on FLAN-Alpaca | 85.31 |

## 6.2 Using Different Vision Features

Different vision features may affect the model performance. We compare three widely-used types of vision features, ViT (Dosovitskiy et al., 2021b), CLIP (Radford et al., 2021), DETR (Carion et al., 2020), and ResNet (He et al., 2016). ViT, CLIP, and DETR are patch-like features. For the ResNet features, we repeat the pooled features of ResNet-50 to the same length with the text sequence to imitate the patch-like features, where each patch is the same as the pooled image features. More details of the vision features are presented in Appendix B.2.

Table 7: Using different vision features.

| Feature | Feature Shape | Accuracy |
|---|---|---|
| ViT | (145, 1024) | 85.31 |
| CLIP | (49, 2048) | 84.27 |
| DETR | (100, 256) | 83.16 |
| ResNet | (512, 2048) | 82.86 |

Table 7 shows the comparative results of vision features. We observe that ViT achieves relatively better performance. Therefore, we use ViT by default in Multimodal-CoT.

## 6.3 Error Analysis

To gain deeper insights into the behavior of Multimodal-CoT and facilitate future research, we manually analyzed randomly selected examples generated by our approach. The categorization results are illustrated in Figure 5. We examined 50 samples that yielded incorrect answers and categorized them accordingly. The examples from each category can be found in Appendix D.

The most prevalent error type is commonsense mistakes, accounting for 80% of the errors. These mistakes occur when the model is faced with questions that require commonsense knowledge, such as interpreting maps, counting objects in images, or utilizing the alphabet. The second error type is logical mistakes, constituting 14% of the errors, which involve contradictions in the reasoning process. Additionally, we have observed cases where incorrect answers are provided despite the CoT being either empty or correct, amounting to 6% of the errors. The CoT in these cases may not necessarily influence the final answer.

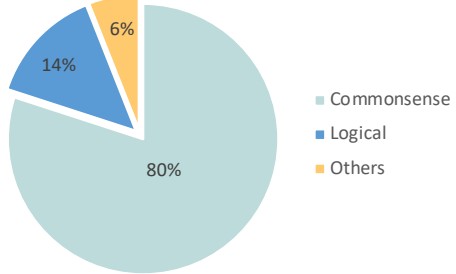

Figure 5: Categorization analysis.

The analysis reveals potential avenues for future research. Enhancements can be made to Multimodal-CoT by: (i) integrating more informative visual features and strengthening the interaction between language and vision to enable comprehension of maps and numerical counting; (ii) incorporating commonsense knowledge; and (iii) implementing a filtering mechanism, such as using only relevant CoTs to infer answers and disregarding irrelevant ones.

## 7 Conclusion

We formally study the problem of multimodal CoT. We propose Multimodal-CoT that incorporates language and vision modalities into a two-stage framework that separates rationale generation and answer inference, so answer inference can leverage better generated rationales from multimodal information. With Multimodal-CoT, our model under 1 billion parameters achieves new state-of-the-art performance on the ScienceQA benchmark. Analysis shows that Multimodal-CoT has the merits of mitigating hallucination and enhancing convergence speed. Our error analysis identifies the potential to leverage more effective vision features, inject commonsense knowledge, and apply filtering mechanisms to improve CoT reasoning in future studies.

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

# A    EXTENDED ANALYSIS FOR THE CHALLENGE OF MULTIMODAL-CoT

## A.1    ADDITIONAL EXAMPLES OF MISLEADING THROUGH HALLUCINATED RATIONALES

Based on our case studies (Section 3.2), we have observed a tendency for the baseline model to generate hallucinated rationales. Here, we present additional examples to illustrate this phenomenon, as depicted in Figure 6.

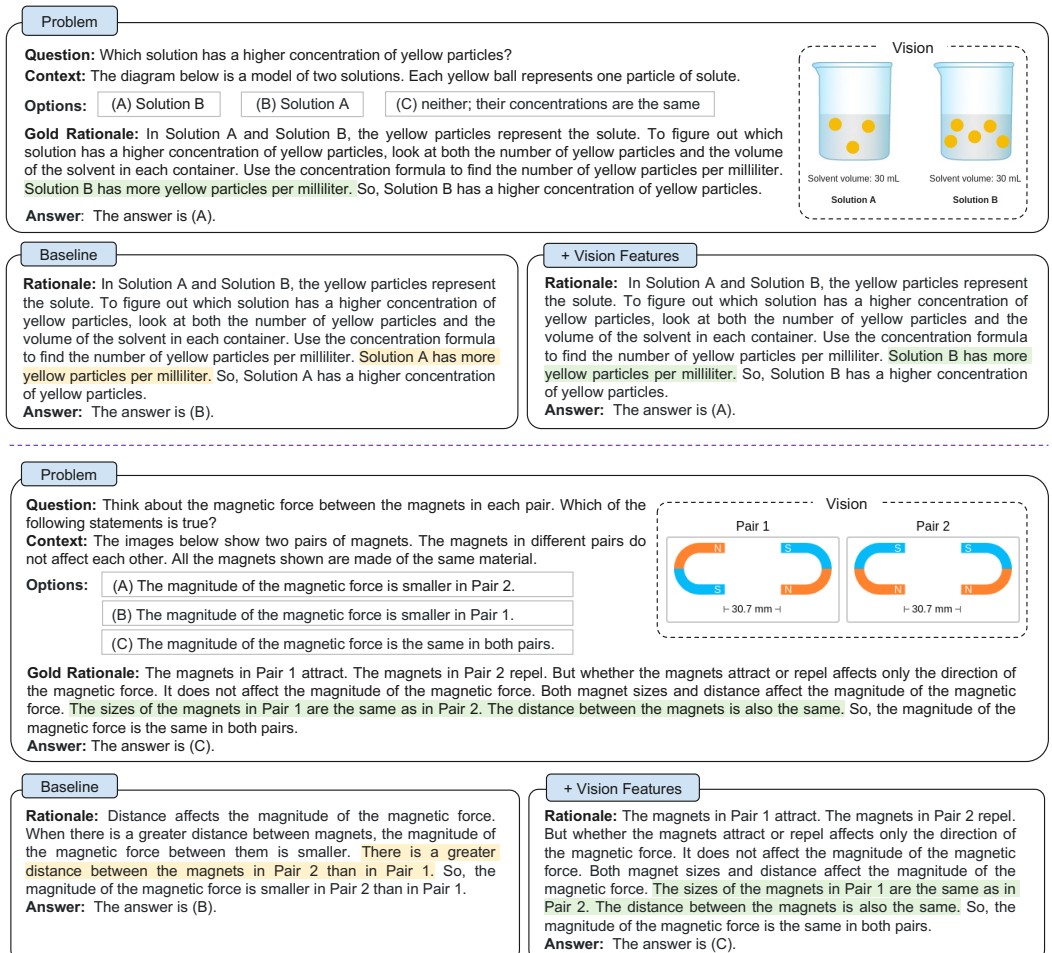

Figure 6: Examples of the two-stage framework without vision features (baseline) and with vision features (ours) for generating rationales and predicting answers. The upper part presents the problem details, and the lower part shows the outputs of the baseline and our method.

## A.2    TWO-STAGE TRAINING PERFORMANCE WITH DIFFERENT SIZES OF LMs

In Section 3, we observed that the inclusion of vision features has a positive impact on the generation of more effective rationales, consequently resulting in improved answer accuracy. In addition to incorporating vision features, another approach to addressing the issue of incorrect rationales is to scale the size of the language model (LM). Figure 7 showcases the answer accuracy achieved by our two-stage training framework, both with and without the integration of vision features. Notably, when employing a larger LM, the baseline accuracy (without vision features) experiences a significant enhancement. This finding suggests that scaling the LM size could potentially alleviate the problem of incorrect rationales. However, it is crucial to acknowledge that the performance still falls considerably short of utilizing vision features. This outcome further validates the effectiveness of our Multimodal-CoT methodology across varying LM sizes.

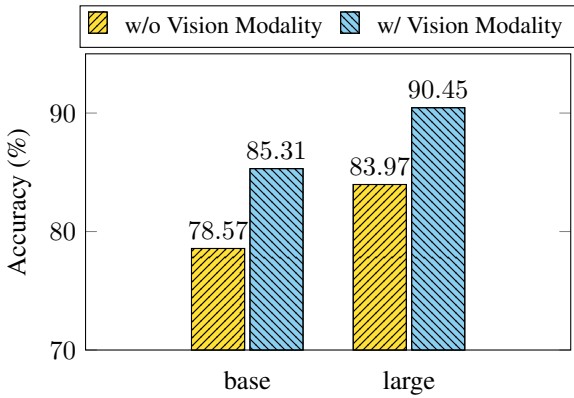

Figure 7: Answer accuracy with different sizes of LMs.

## A.3 DISCUSSION OF THE POSSIBLE PARADIGMS TO ACHIEVE MULTIMODAL-COT

As discussed in Section 1, there are two primary approaches to facilitate Multimodal-CoT reasoning: (i) prompting LLMs and (ii) fine-tuning small models. The common approach in the first approach is to unify the input from different modalities and prompt LLMs to perform reasoning (Zhang et al., 2023a; Lu et al., 2023; Liu et al., 2023; Alayrac et al., 2022; Hao et al., 2022; Yasunaga et al., 2022). For instance, one way to achieve this is by extracting the caption of an image using a captioning model and then concatenating the caption with the original language input to feed LLMs. By doing so, visual information is conveyed to LLMs as text, effectively bridging the gap between modalities. This approach can be represented as the input-output format ¡image → caption, question + caption → answer¿. We refer to this approach as **Caption-based Reasoning** (Figure 8a). It is worth noting that the effectiveness of this approach depends on the quality of the image caption, which may be susceptible to errors introduced during the transfer from image captioning to answer inference.

In contrast, an intriguing aspect of CoT is the ability to decompose complex problems into a series of simpler problems and solve them step by step. This transformation leads to a modification of the standard format ¡question → answer¿ into ¡question → rationale → answer¿. Rationales, being more likely to reflect the reasoning processes leading to the answer, play a crucial role in this paradigm. Consequently, we refer to approaches following this paradigm as **CoT-based Reasoning**. The nomenclature has been widely adopted in the literature (Huang & Chang, 2022; Zhang et al., 2023c; Lu et al., 2022c).

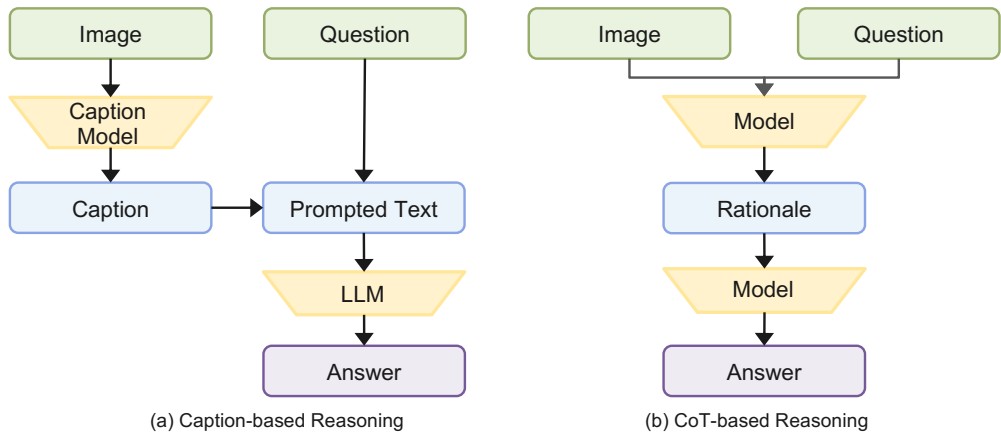

Figure 8: Paradigms to achieve Multimodal-CoT.

Our work aligns with the paradigms of **CoT-based Reasoning** in the context of multimodal scenarios, specifically employing the ¡question + image → rationale → answer¿ framework (Figure 8b). This approach confers advantages on two fronts. Firstly, the Multimodal-CoT framework leverages feature-level interactions between vision and language inputs, enabling the model to gain a deeper understanding of the input information and facilitating more effective inference of answers by incorporating well-founded rationales. Our analysis has demonstrated that Multimodal-CoT offers notable benefits by mitigating hallucination and enhancing convergence, resulting in superior performance on our benchmark datasets. Secondly, the lightweight nature of Multimodal-CoT renders it compatible with resource constraints and circumvents any potential paywalls.

## B    EXPERIMENTAL DETAILS

### B.1    BASELINE METHODS

We utilized three categories of methods as our baselines:

(i) Visual question answering (VQA) models, including MCAN (Yu et al., 2019), Top-Down (Anderson et al., 2018), BAN (Kim et al., 2018), DFAF (Gao et al., 2019), ViLT (Kim et al., 2021), Patch-TRM (Lu et al., 2021), and VisualBERT (Li et al., 2019). These VQA baselines take the question, context, and choices as textual input, while utilizing the image as visual input. They employ a linear classifier to predict the score distribution over the choice candidates.

(ii) LMs, including the text-to-text UnifiedQA model (Khashabi et al., 2020) and few-shot learning LLMs (GPT-3.5, ChatGPT, GPT-4, and Chameleon (Lu et al., 2023)). UnifiedQA (Khashabi et al., 2020) is adopted as it is the best fine-tuning model in Lu et al. (2022a). UnifiedQA takes the textual information as the input and outputs the answer choice. The image is converted into a caption extracted by an image captioning model following Lu et al. (2022a). UnifiedQA treats our task as a text generation problem. In Lu et al. (2022a), it is trained to generate a target answer text, i.e., one of the candidate options. Then, the most similar option is selected as the final prediction to evaluate the question answering accuracy. For GPT-3.5 models (Chen et al., 2020), we use the text-davinci-002 and text-davinci-003 engines due to their strong performance. In addition, we also include the comparison with ChatGPT and GPT-4. The inference is based on the few-shot prompting, where two in-context examples from the training set are concatenated before the test instance. The few-shot demonstrations are the same as those in Lu et al. (2022a).

(iii) Fine-tuned large vision-language model. We select the recently released LLaMA-Adapter (Zhang et al., 2023a), LLaVA (Liu et al., 2023), and InstructBLIP (Dai et al., 2023) as the competitive large vision-language baselines. The backbone model is the 7B LLaMA model fine-tuned with $52k$ self-instruct demonstrations. To adapt to our tasks, the model is further fine-tuned on the ScienceQA dataset.

For UnifiedQA and GPT-family models, CoT is applied after the answer (Lu et al., 2022a). Besides the above baselines, we develop a stronger baseline by slightly modifying the output format of UnifiedQA. Instead of predicting the answer texts, our baseline directly predicts the choice, e.g., *the answer is B*. This setting helps our baseline achieve better results than the existing UnifiedQA. Therefore, we use the stronger method as the language-only baseline for analysis.

### B.2    DETAILS OF VISION FEATURES

In Section 6.2, we compared four types of vision features, ViT (Dosovitskiy et al., 2021b), CLIP (Radford et al., 2021), DETR (Carion et al., 2020), and ResNet (He et al., 2016). The specific models are: (i) ViT: *vit_large_patch32_384*,[7] (ii) CLIP: RN101;[8] (iii) DETR: *detr_resnet101_dc5*;[9] (iv) ResNet: we use the averaged pooled features of a pre-trained ResNet50 CNN.

Table 8 presents the dimension of the vision features (after the function VisionExtractor($\cdot$) in Eq. 3). For ResNet-50, we repeat the pooled features of ResNet-50 to the same length as the text sequence to imitate the patch-like features, where each patch is the same as the pooled image features.

---

[7]`https://github.com/rwightman/pytorch-image-models`.
[8]`https://github.com/jianjieluo/OpenAI-CLIP-Feature`.
[9]`https://github.com/facebookresearch/detr`.

Table 8: Feature shape of vision features

| Method | Feature Shape |
|--------|---------------|
| ViT    | (145, 1024)   |
| CLIP   | (49, 2048)    |
| DETR   | (100, 256)    |
| ResNet | (512, 2048)   |

### B.3 DATASETS

Our method is evaluated on the ScienceQA (Lu et al., 2022a) and A-OKVQA (Schwenk et al., 2022) benchmark datasets.

• ScienceQA is a large-scale multimodal science question dataset with annotated lectures and explanations. It contains $21k$ multimodal multiple choice questions with rich domain diversity across 3 subjects, 26 topics, 127 categories, and 379 skills. The dataset is split into training, validation, and test splits with $12k$, $4k$, and $4k$ questions, respectively.

• A-OKVQA is a knowledge-based visual question answering benchmark, which has $25k$ questions requiring a broad base of commonsense and world knowledge to answer. Each question is annotated with rationales that explain why a particular answer was correct according to necessary facts or knowledge. It has $17k/1k/6k$ questions for train/val/test.

For ScienceQA, our model is evaluated on the test set. For A-OKVQA, our model is evaluated on the validation set as the test set is hidden.

### B.4 IMPLEMENTATION DETAILS OF MULTIMODAL-CoT

As the Multimodal-CoT task requires generating the reasoning chains and leveraging the vision features, we adopt the T5 encoder-decoder architecture (Raffel et al., 2020) under `Base` (200M) and `large` (700M) settings in our framework. We apply FLAN-Alpaca to initialize our model weights.[10] We will show that Multimodal-CoT is generally effective with other backbone LMs, such as UnifiedQA (Khashabi et al., 2020) and FLAN-T5 (Chung et al., 2022) (Section 6.1). The vision features are obtained by the frozen ViT-large encoder (Dosovitskiy et al., 2021b). Since using image captions can slightly improve model performance, as shown in Section 3.3, we append the image captions to the context following Lu et al. (2022a). The captions are generated by InstructBLIP (Dai et al., 2023). We fine-tune the models up to 20 epochs, with a learning rate selected in {5e-5, 8e-5}. The maximum input sequence lengths for rationale generation and answer inference are 512 and 64, respectively. The batch size is 8. Our experiments are run on 8 NVIDIA Tesla V100 32G GPUs.

## C FURTHER ANALYSIS

### C.1 ABLATION STUDY

Ablation study results in Table 9 show that both the integration of vision features and the two-stage framework design contribute to the overall performance. These findings provide strong evidence for the effectiveness of multimodality and highlight the potential for achieving CoT reasoning using 1B-models through our proposed two-stage framework.

Table 9: Ablation results of Multimodal-CoT.

| Model | Base | Large |
|-------|------|-------|
| Multimodal-CoT | 85.31 | 90.45 |
| w/o Two-Stage Framework | 82.62 | 84.56 |
| w/o Vision Features | 78.57 | 83.97 |

---

[10]`https://github.com/declare-lab/flan-alpaca.`

## C.2 Multimodality Boosts Convergence

Figure 9 shows the validation accuracy curve of the baseline and Multimodal-CoT across different training epochs. "One-stage" is based on the QCM→A input-output format as it achieves the best performance in Table 2 and "Two-stage" is our two-stage framework. We find that the two-stage methods achieve relatively higher accuracy at the beginning than the one-stage baselines that generate the answer directly without CoT. However, without the vision features, the two-stage baseline could not yield better results as the training goes on due to the low-quality rationales (as observed in Section 3). In contrast, using vision features helps generate more effective rationales that contribute to better answer accuracy in our two-stage multimodal variant.

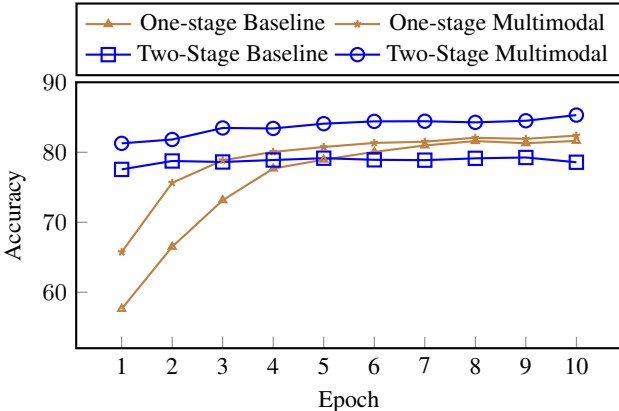

Figure 9: Accuracy curve of the No-CoT baseline and Multimodal-CoT variants.

## C.3 When Multimodal-CoT Meets Large Models

A recent flame is to leverage large language models or large vision-language models to generate reasoning chains for multimodal question answering problems (Zhang et al., 2023a; Lu et al., 2023; Liu et al., 2023; Alayrac et al., 2022; Hao et al., 2022; Yasunaga et al., 2022). We are interested in whether we can use large models to generate the rationales for Multimodal-CoT; thus breaking the need for datasets with human-annotated rationales. During the first-stage training of Multimodal-CoT, our target rationales are based on human annotation in the benchmark datasets. Now, we replace the target rationales with those generated by an LLM or a vision-language model. Concretely, we feed the questions with images (IMG) and the question without images (TXT) to InstructBLIP (Dai et al., 2023) (Figure 10a) and ChatGPT (Figure 10b) for zero-shot inference, respectively. Then, we use the generated pseudo-rationales as the target rationales for training instead of relying on the human annotation of reasoning chains.

Table 10 shows the comparison results. We see that using the generated rationales achieves comparable performance to using human-annotated rationales for training. In addition, the performance is also much better than directly prompting those baseline models to obtain the answer (in the QCM→A inference format).

Table 10: Result comparison with large models. We also present the results of InstructBLIP and ChatGPT baselines for reference. The inference format for the two baselines is QCM→A.

| Model | IMG | TXT | AVG |
|---|---|---|---|
| InstructBLIP | 60.50 | - | - |
| ChatGPT | 56.52 | 67.16 | 65.95 |
| Multimodal-CoT w/ Annotation | 88.25 | 90.13 | 90.45 |
| Multimodal-CoT w/ Generation | 83.54 | 85.73 | 87.76 |

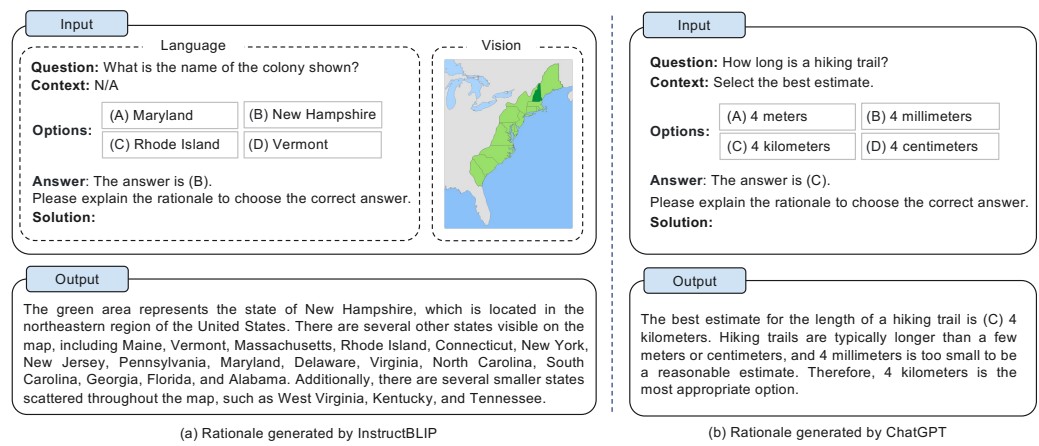

Figure 10: Rationale generation examples

We see that Multimodal-CoT can work effectively with large models. The findings above compellingly show the feasibility of adaptation to scenarios without human-annotated rationales, thereby establishing the effectiveness of our approach across diverse tasks.

## C.4 ALIGNMENT STRATEGIES FOR MULTIMODAL INTERACTION

We are interested in whether using different alignment strategies for multimodal interaction may contribute to different behaviors of multimodal-CoT. To this end, we tried another alignment strategy, i.e., image-grounded text encoder, in BLIP Li et al. (2022b). This alignment approach injects visual information by inserting one additional cross-attention layer between the self-attention layer and the feed-forward network for each transformer block of the text encoder. Our current strategy in the paper is similar to the unimodal encoder as in BLIP, which is used for comparison.

Table 11: Result comparison with different alignment strategies for multimodal interaction.

| Model | Accuracy |
|---|---|
| Direct Answering | 82.62 |
| Unimodal encoder | 85.31 |
| Image-grounded text encoder | 84.60 |

In Table 11, we see that using other alignment strategies also contributes to better performance than direct answering.

## C.5 DETAILED RESULTS OF MULTIMODAL-COT ON DIFFERENT BACKBONE MODELS

To test the generality of the benefits of our approach to other backbone models, we alter the underlying LMs to other variants of different types. As detailed results shown in Table 12, our approach is generally effective for the widely used backbone models.

Table 12: Detailed results of Multimodal-CoT on different backbone models.

| Model | NAT | SOC | LAN | TXT | IMG | NO | G1-6 | G7-12 | Avg |
|---|---|---|---|---|---|---|---|---|---|
| MM-CoT on UnifiedQA | 80.60 | 89.43 | 81.00 | 80.50 | 80.61 | 81.74 | 82.38 | 82.86 | 82.55 |
| MM-CoT on FLAN-T5 | 81.39 | 90.89 | 80.64 | 80.79 | 80.47 | 82.58 | 83.48 | 82.66 | 83.19 |
| MM-CoT on FLAN-Alpaca | 84.06 | 92.35 | 82.18 | 82.75 | 82.75 | 84.74 | 85.79 | 84.44 | 85.31 |

## D    EXAMPLES OF CASE STUDIES

To gain deeper insights into the behavior of Multimodal-CoT and facilitate future research, we manually analyzed randomly selected examples generated by our approach. The categorization results are illustrated in Figure 11. We examined 50 samples that yielded incorrect answers and categorized them accordingly.

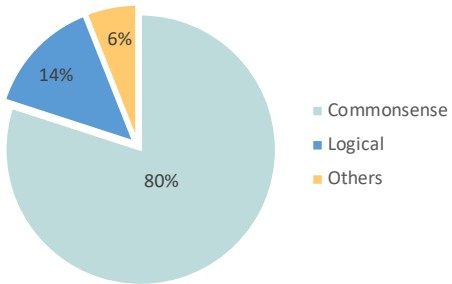

Figure 11: Categorization analysis.

The most prevalent error type is commonsense mistakes, accounting for 80% of the errors. These mistakes occur when the model is faced with questions that require commonsense knowledge, such as interpreting maps (Figure 12a), counting objects in images (Figure 12b), or utilizing the alphabet (Figure 12c).

The second error type is logical mistakes, constituting 14% of the errors, which involve comparison mistakes (Figure 13a) and contradictions in the reasoning process (Figure 13b).

Additionally, we have observed cases where incorrect answers are provided despite the CoT being either empty or correct, amounting to 6% of the errors. The CoT in these cases may not necessarily influence the final answer (Figure 14).

The analysis reveals potential avenues for future research. Enhancements can be made to Multimodal-CoT by: (i) integrating more informative visual features and strengthening the interaction between language and vision to enable comprehension of maps and numerical counting; (ii) incorporating commonsense knowledge; and (iii) implementing a filtering mechanism, such as using only relevant CoTs to infer answers and disregarding irrelevant ones.

**Problem**

**Question:** Which of these cities is marked on the map?

**Context:** N/A

**Options:** (A) Las Vegas   (B) San Francisco   (C) Phoenix   (D) Salt Lake City

**Rationale :** The city is Las Vegas, Nevada. Salt Lake City, Phoenix, and San Francisco are marked with gray circles on the map below.
**Answer:** The answer is (A).

Vision

**Prediction**

**Rationale:** The city is San Francisco, California. Salt Lake City, Las Vegas, and Phoenix are marked with gray circles on the map below.
**Answer:** The answer is (B).

(a) Mistakes about maps.

**Problem**

**Question:** Complete the text to describe the diagram. Solute particles moved in both directions across the permeable membrane. But more solute particles moved across the membrane (). When there was an equal concentration on both sides, the particles reached equilibrium.

**Context:** The diagram below shows a solution with one solute. Each solute particle is represented by a green ball. The solution fills a closed container that is divided in half by a membrane. The membrane, represented by a dotted line, is permeable to the solute particles. The diagram shows how the solution can change over time during the process of diffusion.

**Options:** (A) to the left than to the right   (B) to the right than to the left

**Rationale :** (*extract*) It shows you how the solution changed during the process of diffusion. Before the solute particles reached equilibrium, there were 3 solute particles on the left side of the membrane and 5 solute particles on the right side of the membrane. When the solute particles reached equilibrium, there were 4 solute particles on each side of the membrane. There was 1 more solute particle on the left side of the membrane than before. So, for the solute particles to reach equilibrium, more solute particles must have moved across the membrane to the left than to the right.
**Answer:** The answer is (A).

Vision

**Prediction**

**Rationale:** (*extract*) It shows you how the solution changed during the process of diffusion. Before the solute particles reached equilibrium, there were 5 solute particles on the left side of the membrane and 4 solute particles on the right side of the membrane. When the solute particles reached equilibrium, there were 5 solute particles on each side of the membrane. There was 1 more solute particle on the right side of the membrane than before. So, for the solute particles to reach equilibrium, more solute particles must have moved across the membrane to the right than to the right.
**Answer:** The answer is (B).

(b) Mistakes about counting numbers in the image.

**Problem**

**Question:** Which word would you find on a dictionary page with the following guide words? helping - hunter

**Context:** The diagram below shows a solution with one solute. Each solute particle is represented by a green ball. The solution fills a closed container that is divided in half by a membrane. The membrane, represented by a dotted line, is permeable to the solute particles. The diagram shows how the solution can change over time during the process of diffusion.

**Options:** (A) hospital   (B) hardship

**Rationale :** Put the words in alphabetical order. Since hospital is between the guide words helping - hunter, it would be found on that page.
**Answer:** The answer is (A).

Vision

N/A

**Prediction**

**Rationale:** Put the words in alphabetical order. Since hardship is between the guide words helping - hunter, it would be found on that page.
**Answer:** The answer is (B).

(c) Mistakes about utilizing alphabet.

Figure 12: Examples of commonsense mistakes.

---

**Problem**

**Question:** Compare the motion of two cars. Which car was moving at a lower speed?
**Context:** N/A

**Vision**
N/A

**Options:** (A) a car that moved 445miles in 10hours    (B) a car that moved 170miles in 10hours

**Rationale :** Look at the distance each car moved and the time it took to move that distance. One car moved 170 miles in 10 hours. The other car moved 445 miles in 10 hours. Notice that each car spent the same amount of time moving. The car that moved 170 miles moved a shorter distance in that time. So, that car must have moved at a lower speed.
**Answer:** The answer is (B).

**Prediction**

**Rationale:** Look at the distance each car moved and the time it took to move that distance. One car moved 445 miles in 10 hours. The other car moved 170 miles in 10 hours. Notice that each car spent the same amount of time moving. The car that moved 445 miles moved a shorter distance in that time. So, that car must have moved at a lower speed.
**Answer:** The answer is (A).

(a) Logical mistake where the model fails at comparisons.

**Problem**

**Question:** Based on this information, what is Sage's phenotype for the coat color trait?
**Context:** In a group of horses, some individuals have a black coat and others have a reddish-brown coat. In this group, the gene for the coat color trait has two alleles. The allele for a reddish-brown coat (l) is recessive to the allele for a black coat (L). Sage is a horse from this group. Sage has the homozygous genotype ll for the coat color gene.

**Vision**
N/A

**Options:** (A) a black coat    (B) a reddish-brown coat

**Rationale :** Sage's genotype for the coat color gene is ll. Sage's genotype of ll has only l alleles. The l allele is for a reddish-brown coat. So, Sage's phenotype for the coat color trait must be a reddish-brown coat. To check this answer, consider whether Sage's alleles are dominant or recessive. The allele for a reddish-brown coat (l) is recessive to the allele for a black coat (L). This means L is a dominant allele, and l is a recessive allele. Sage's genotype of ll has only recessive alleles. An organism with only recessive alleles for a gene will have the recessive allele's version of the trait. So, Sage's phenotype for the coat color trait must be a reddish-brown coat.
**Answer:** The answer is (B).

**Prediction**

**Rationale:** Sage's genotype for the coat color gene is ll. Sage's genotype of ll has only l alleles. The l allele is for a reddish-brown coat. So, Sage's phenotype for the coat color trait must be a reddish-brown coat. To check this answer, consider whether Sage's alleles are dominant or recessive. The allele for a reddish-brown coat (l) is recessive to the allele for a black coat (L). This means L is a dominant allele, and l is a recessive allele. Sage's genotype of ll has only recessive alleles. An organism with only recessive alleles for a gene will have the recessive allele's version of the trait. So, Sage's phenotype for the coat color trait must be a black coat.
**Answer:** The answer is (A).

(b) Logical mistake where the there is a contradiction in the rationale.

Figure 13: Examples of logical mistakes.

---

**Problem**

**Question:** Which trait did Palaeopython have? Select the trait you can observe on the fossil.
**Context:** This picture shows a fossil of an ancient animal called Palaeopython. Palaeopython lived in trees and could grow more than six feet long.

**Vision**
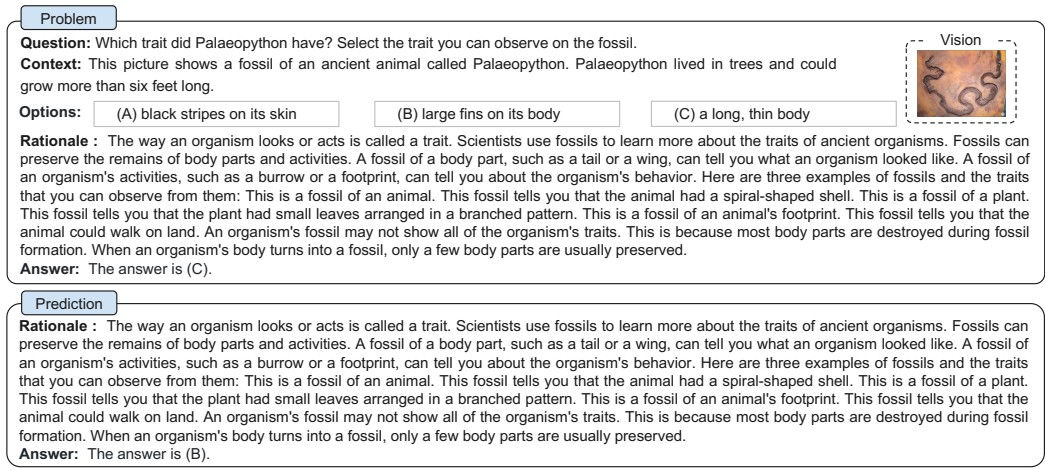

**Options:** (A) black stripes on its skin    (B) large fins on its body    (C) a long, thin body

**Rationale :** The way an organism looks or acts is called a trait. Scientists use fossils to learn more about the traits of ancient organisms. Fossils can preserve the remains of body parts and activities. A fossil of a body part, such as a tail or a wing, can tell you what an organism looked like. A fossil of an organism's activities, such as a burrow or a footprint, can tell you about the organism's behavior. Here are three examples of fossils and the traits that you can observe from them: This is a fossil of an animal. This fossil tells you that the animal had a spiral-shaped shell. This is a fossil of a plant. This fossil tells you that the plant had small leaves arranged in a branched pattern. This is a fossil of an animal's footprint. This fossil tells you that the animal could walk on land. An organism's fossil may not show all of the organism's traits. This is because most body parts are destroyed during fossil formation. When an organism's body turns into a fossil, only a few body parts are usually preserved.
**Answer:** The answer is (C).

**Prediction**

**Rationale :** The way an organism looks or acts is called a trait. Scientists use fossils to learn more about the traits of ancient organisms. Fossils can preserve the remains of body parts and activities. A fossil of a body part, such as a tail or a wing, can tell you what an organism looked like. A fossil of an organism's activities, such as a burrow or a footprint, can tell you about the organism's behavior. Here are three examples of fossils and the traits that you can observe from them: This is a fossil of an animal. This fossil tells you that the animal had a spiral-shaped shell. This is a fossil of a plant. This fossil tells you that the plant had small leaves arranged in a branched pattern. This is a fossil of an animal's footprint. This fossil tells you that the animal could walk on land. An organism's fossil may not show all of the organism's traits. This is because most body parts are destroyed during fossil formation. When an organism's body turns into a fossil, only a few body parts are usually preserved.
**Answer:** The answer is (B).

Figure 14: Examples of answers are incorrect while the CoT is correct.

