# OpenReview forum: "Multimodal Chain-of-Thought Reasoning in Language Models"
_ICLR.cc/2024/Conference — Submitted to ICLR 2024_

### Official Review · Reviewer_mQUb · 2023-11-01

**Soundness:** 3 good
**Presentation:** 3 good
**Contribution:** 2 fair
**Rating:** 5
**Confidence:** 3

**Summary:**

The paper introduces an approach titled "Multimodal-CoT," which focuses on the amalgamation of both language (text) and vision (images) modalities in a two-stage framework. The first stage addresses rationale generation, while the second concentrates on answer inference. An important feature of this paper is its strategy of integrating vision features into the language model. Such a technique, though previously explored in multimodal language and vision models, is shown to decrease rationale hallucination and consequently boost answer accuracy.

**Strengths:**

1. **Quality:** The results seem promising, especially given the observation that this approach meshes well with various backbone models.

2. **Clarity:** The paper appears to provide comprehensive baselines and analyses, suggesting that the methodology and results have been presented in a clear and structured manner.

3. **Significance:** The ablation study emphasizes the significance of both the integration of vision features and the two-stage framework. These results suggest that each component of the design distinctly contributes to the observed enhancement in performance.

**Weaknesses:**

1. **Incremental Novelty:** The work leans heavily on prior multimodal LLM research (e.g., BLIP-2 (https://arxiv.org/pdf/2301.12597.pdf), MINIGPT-4 (https://arxiv.org/pdf/2304.10592.pdf)), specifically in terms of incorporating vision features into the language model. This reduces the perceived novelty of the presented model.

2. **Framework Design:** The two-stage framework, though effective, comes across as relatively straightforward. A deeper exploration or the introduction of more intricate strategies could potentially lead to even more enhanced results.

**Questions:**

1. **Differentiation from CoT:** It might not be accurate to call the proposed approach a variant of CoT, instead it's more like a two-stage pipeline framework. Additionally, what are the potential benefits or shortcomings of adopting a "QCM->RA" strategy, especially when paired with few-shot demonstrations? This alternative is more like a standard CoT approach. The authors are encouraged to compare with this baseline.

2. **Additional Benchmark Results:** For a holistic understanding, it would be beneficial to see the results of "UnifiedQA," "FLAN-T5," and "FLAN-Alpaca" in Table 6. This would provide a comprehensive view and easy comparison with existing models.

---

> ### Author Response · Authors · 2023-11-19
>
> Thanks for your insightful review and constructive feedback.
>
> > W1: Incremental Novelty.
>
> Compared with previous techniques, our work is in sharp contrast with the following three aspects.
>
> Firstly, this work is the first to study CoT reasoning in different modalities among existing publications.
>
> Secondly, as a pioneer and effective approach to deal with the multimodal CoT problem, our technique is different from either previous text-only reasoning or VQA approaches by providing a means of explicit intermediate reasoning with multimodal input powered by a frozen image encoder and two-stage framework.
>
> Thirdly, our two-stage approach elicits the analysis of why the naive way of employing CoT fails in the context and how incorporating vision features alleviates the problem. The approach has been shown generally effective across tasks and backbone models. Furthermore, we conduct in-depth studies to investigate the benefits and limitations of Multimodal-CoT to facilitate future studies.
>
> > W2: Framework Design.
>
> We explore the framework design to gain insights in the following ways.
>
> Firstly, we have tried intricate alignment strategies inspired by BLIP, i.e., Image-grounded text encoder according to Section 3.1 in BLIP. This alignment approach injects visual information by inserting one additional cross-attention layer between the self-attention layer and the feed-forward network for each transformer block of the text encoder. Our current strategy in the paper is similar to the Unimodal encoder as in BLIP, which is used for comparison. We see that using other alignment strategies also contributes to better performance than direct answering.
>
> | Method | Accuracy |
> | ---------- | ---------------- |
> | Direct Answering | 82.62 |
> | Unimodal encoder | 85.31 |
> | Image-grounded text encoder | 84.60  |
>
> Secondly, we explore the possibilities of intexegrating Multimodal-CoT with more complex models. We find that our method is orthogonal to existing multimodal models (e.g., InstructBLIP) and can be potentially used with them together to further improve generality. We have provided a detailed discussion in Appendix C.3.
>
> > Q1: Differentiation from CoT.
>
> Great point. Our preliminary motivation is to adopt QCM->RA. However, the performance is much inferior than direct reasoning (QCM->A) as shown in Table 2. Our observation is that the "X->RA" strategy (we replace "QCM->RA" to "X->RA" for the sake of generality) is beneficial when the model can generate effective rationales, e.g., the standard CoT approach in complex reasoning tasks such as arithmetic, commonsense, and logical reasoning tasks.
>
> However, we find that the rationales in our tasks often contain commonse or logical mistakes according to our error analysis (similar to our case studies in Appendix D, as a reference). When using the rationales in the output side (RA), the decoder would generates the answer conditioned on the previously generated rationales. In this way, the answer generation may more likely suffer from the imperfect rationales. In contrast, when taking the rationales in the input side, the rationale will not directly affect the answer generation process of the decoder and the model may also easily ignore the imperfect rationales.
>
> This finding also applies to the in-context learning baselines with few-shot demonstrations. According to the results in [1], GPT-3 also yields inferior performance with X->RA.
>
> | Method | Accuracy |
> | ---------- | ---------------- |
> | X -> A | 74.0 |
> | X -> AR | 73.6 |
> | X -> RA | 55.4 |
>
> Therefore, we finally adopt the two-stage framework and finally succeeds in improved performance.
>
> > Q2: Additional Benchmark Results.
>
> Yes. We have added the detailed benchmark results in Appendix C5.
>
> Reference:
>
> [1] Pan Lu, Swaroop Mishra, Tony Xia, Liang Qiu, Kai-Wei Chang, Song-Chun Zhu, Oyvind Tafjord, Peter Clark, and Ashwin Kalyan. Learn to explain: Multimodal reasoning via thought chains for science question answering. Advances in Neural Information Processing Systems, 35, 2507-2521.

---

> ### Comment · Reviewer_mQUb · 2023-12-01
>
> Thank the authors for the response! I still have a few questions:
> 1) "This finding also applies to the in-context learning baselines with few-shot demonstrations." Are the new results mentioned in your response from paper [1] or your experiments? Can you report the few-shot demonstrations results in your experiments?
> 2) The detailed benchmark results in Appendix C5 didn't report the results of baseline backbones: "UnifiedQA," "FLAN-T5," and "FLAN-Alpaca".
>
> [1] Pan Lu, Swaroop Mishra, Tony Xia, Liang Qiu, Kai-Wei Chang, Song-Chun Zhu, Oyvind Tafjord, Peter Clark, and Ashwin Kalyan. Learn to explain: Multimodal reasoning via thought chains for science question answering. Advances in Neural Information Processing Systems, 35, 2507-2521.

---

### Official Review · Reviewer_9gAH · 2023-11-01

**Soundness:** 4 excellent
**Presentation:** 4 excellent
**Contribution:** 4 excellent
**Rating:** 5
**Confidence:** 3

**Summary:**

The paper introduces a novel approach that integrates language and vision modalities into a two-stage framework for large language models. This framework enhances reasoning by generating intermediate reasoning chains before inferring answers. The method shows new state-of-the-art performance on the ScienceQA benchmark, particularly effective in models under 1 billion parameters, and addresses the issue of hallucination in answer inference.

**Strengths:**

1. Innovative Approach: The integration of multimodal data (text and images) into CoT reasoning is a significant advancement, addressing a gap in previous research which focused mainly on language modality.
2. Mitigation of Hallucination: The approach specifically targets and successfully mitigates the issue of hallucination in answer inference, a common problem in smaller language models.
3. Detailed Analysis: The paper provides a comprehensive background study and analysis of existing CoT techniques, enhancing the understanding of the field.

**Weaknesses:**

1. Limited Scope of Evaluation: The paper only evaluated their approach using 2 benchmark datasets like ScienceQA and AOKVQA. While these datasets are relevant and challenging, the paper represents a specific type of reasoning tasks.
2. The paper demonstrates the effectiveness of the proposed method primarily in the context of encoder-decoder models. However, its effectiveness in popular left-to-right language models, which are widely used, is not explicitly addressed. This omission can limit the understanding of how the proposed method might perform or be adapted to these prevalent LMs.

**Questions:**

N/A

---

> ### Author Response · Authors · 2023-11-19
>
> Thanks for your insightful review and constructive feedback.
>
> > W1: "Limited Scope of Evaluation: The paper only evaluated their approach using 2 benchmark datasets like ScienceQA and AOKVQA. While these datasets are relevant and challenging, the paper represents a specific type of reasoning tasks."
>
> We adopt the ScienceQA and A-OKVQA datasets for evaluation because they have annotated rationales as golden labels to assess the quality of the generated rationales and to have an in-depth study on the problem of multimodal CoT, though the annotations are dispensable in real use. Our results compellingly show the feasibility of adaptation to scenarios even without human-annotated rationales, thereby establishing the effectiveness of our approach across diverse tasks.Please also note that ScienceQA is a large-scale multimodal QA benchmark that covers rich domain diversity across 3 subjects, 26 topics, 127 categories, and 379 skills.
>
> Indeed,  these datasets are relevant and challenging. It is worth noting that existing publications only used one of the datasets for evaluation. In contrast, we have taken both of them to verify the effectiveness of Multimodal-CoT [1-5].
>
> > W2: "The paper demonstrates the effectiveness of the proposed method primarily in the context of encoder-decoder models. However, its effectiveness in popular left-to-right language models, which are widely used, is not explicitly addressed. This omission can limit the understanding of how the proposed method might perform or be adapted to these prevalent LMs."
>
> Our choice of encoder-decoder architecture is mostly because of two reasons. Firstly, encoder-decoder architecture is also widely adopted as multimodal framework in this research field. Secondly, existing studies [6] have already compared the performance between encoder-decoder architecture (T5) and left-to-right decoder-only architecture (LLaMA/Vinuca). The results show that T5 achieves better performance on most of the tasks.  In addition, we have adopted various left-to-right language models as baselines presented in Table 4, including LLaMA-Adapter, LLaVA, InstructBLIP, and Chameleon. Results show that our approach, with less than 1B parameters, perform better than most of those latest work.
>
> References:
>
> [1] Liangke Gui, Borui Wang, Qiuyuan Huang, Alex Hauptmann, Yonatan Bisk, Jianfeng Gao. KAT: A Knowledge Augmented Transformer for Vision-and-Language. In Proceedings of the 2022 Conference of the North American Chapter of the Association for Computational Linguistics: Human Language Technologies (pp. 956-968).
>
> [2] Yuanze Lin, Yujia Xie, Dongdong Chen, Yichong Xu, Chenguang Zhu, Lu Yuan. Revive: Regional visual representation matters in knowledge-based visual question answering. Advances in Neural Information Processing Systems, 35, 10560-10571.
>
> [3] Haotian Liu, Chunyuan Li, Qingyang Wu, and Yong Jae Lee. Visual instruction tuning. The Thirty-seventh Conference on Neural Information Processing Systems (NeurIPS 2023).
>
> [4] Pan Lu, Baolin Peng, Hao Cheng, Michel Galley, Kai-Wei Chang, Ying Nian Wu, Song-Chun Zhu, and Jianfeng Gao. Chameleon: Plug-and-play compositional reasoning with large language models. The Thirty-seventh Conference on Neural Information Processing Systems (NeurIPS 2023).
>
> [5] Renrui Zhang, Jiaming Han, Aojun Zhou, Xiangfei Hu, Shilin Yan, Pan Lu, Hongsheng Li, Peng Gao, and Yu Qiao. LLaMA-Adapter: Efficient Fine-tuning of Language Models with Zero-init Attention. arXiv preprint arXiv:2303.16199.
>
> [6] Wenliang Dai, Junnan Li, Dongxu Li, Anthony Meng Huat Tiong, Junqi Zhao, Weisheng Wang, Boyang Li, Pascale Fung, and Steven Hoi. Instructblip: Towards general-purpose vision-language models with instruction tuning, 2023.

---

### Official Review · Reviewer_MqoG · 2023-11-02

**Soundness:** 3 good
**Presentation:** 3 good
**Contribution:** 3 good
**Rating:** 5
**Confidence:** 4

**Summary:**

This paper presents multimodal chain-of-thought. Comprehensive analysis shows that the proposed model outperforms the state-of-the-art models on two benchmarks.

**Strengths:**

- State-of-the-art performance on two benchmarks.
- Simple yet effective approach on improving reasoning in vision and language settings.
- Comprehensive analysis on the proposed model.

**Weaknesses:**

- A few arguments are not convincing or well-supported. For instance, more rigorous experiments are needed to claim *surpassing human performance*: on the one hand, humans can show significant variances when working on the same problem; on the other hand, ScienceQA collects the human performance baseline with Amazon Mechanical Turk, which is quite hard to control the data quality.
- This paper overclaims on multimodal CoT, while only vision and text are evaluated. Other modalities, such as audio, video, and touch, are not supported in the model.
- A few points are not clear enough (see below for details).

**Questions:**

1. It's surprising that a model with a FLAN-Alpaca-Base backbone can outperform GPT-4 (Table 4). Is the GPT-4 you tested multimodal or text-only? Did you use CoT prompting for GPT-4 as well?
2. Related to the above question, shouldn't multimodal-CoT be considered as a prompting technique, which is orthogonal to the base model? If so, the references in Table 4 are probably misleading.

---

> ### Author Response · Authors · 2023-11-19
>
> Thanks for your insightful review and constructive feedback.
>
> > W1: "A few arguments are not convincing or well-supported. For instance, more rigorous experiments are needed to claim surpassing human performance: on the one hand, humans can show significant variances when working on the same problem; on the other hand, ScienceQA collects the human performance baseline with Amazon Mechanical Turk, which is quite hard to control the data quality."
>
> Yes, we agree. We report human performance following the original ScienceQA paper. It can be seen as a piece of evidence to indicate the superior performance of our approach as it is the first to suppose human performance. Due to the lack of details in human evaluations in the ScienceQA paper, we could not elaborate on how the model surpasses human performance in specific artifacts. To avoid confusion, we have removed those arguments.
>
> > W2: "This paper overclaims on multimodal CoT, while only vision and text are evaluated. Other modalities, such as audio, video, and touch, are not supported in the model."
>
> We call Multimodal basically following a wide range of existing studies that focus on vision and language modalities and the framework is generally applicable to other modalities. Following the comment, we have added a clarification in Footnote 1.
>
> > W3: "A few points are not clear enough (see below for details)."
>
> - Q1: It's surprising that a model with a FLAN-Alpaca-Base backbone can outperform GPT-4 (Table 4). Is the GPT-4 you tested multimodal or text-only? Did you use CoT prompting for GPT-4 as well?
>
> It is a text-only GPT-4. It is based on CoT prompting according to [1].
>
> - Q2: Related to the above question, shouldn't multimodal-CoT be considered as a prompting technique, which is orthogonal to the base model? If so, the references in Table 4 are probably misleading.
>
> We are afraid not. Multimodal-CoT is a fine-tuned model. As we discussed in Section 1, prompting and fine-tuning are two potential solutions for Multimodal-CoT. The prompting results are reported for reference. In contrast, fine-tuning approaches generally achieve better performance and are widely adopted by related studies (those latest baselines).
> Therefore, we choose the fine-tuning paradigm to design Multimodal-CoT and focus on the compairon with fine-tuning approaches.
>
> [1] Pan Lu, Swaroop Mishra, Tony Xia, Liang Qiu, Kai-Wei Chang, Song-Chun Zhu, Oyvind Tafjord, Peter Clark, and Ashwin Kalyan. Learn to explain: Multimodal reasoning via thought chains for science question answering. Advances in Neural Information Processing Systems, 35, 2507-2521.
>
> [2] Michihiro Yasunaga, Armen Aghajanyan, Weijia Shi, Rich James, Jure Leskovec, Percy Liang, Mike Lewis, Luke Zettlemoyer, and Wen-tau Yih. Retrieval-augmented multimodal language modeling. Proceedings of the 40th International Conference on Machine Learning, PMLR 202:39755-39769.
>
> [3] Zhuosheng Zhang, Kehai Chen, Rui Wang, Masao Utiyama, Eiichiro Sumita, Zuchao Li, and Hai Zhao. Universal multimodal representation for language understanding. IEEE Transactions on Pattern Analysis and Machine Intelligence, pp. 1–18, 2023b. doi: 10.1109/TPAMI.2023.3234170.

---

### Official Review · Reviewer_USEM · 2023-11-08

**Soundness:** 3 good
**Presentation:** 3 good
**Contribution:** 3 good
**Rating:** 6
**Confidence:** 4

**Summary:**

This paper proposed a two-stage multimodal CoT framework Multimodal-CoT, which separated rationale generation and answer inference. Through finetuning small models, this paper fused the vision features with the encoded language representations. Multimodal-CoT can alleviate the hallucinations while generating rationales and improving the accuracy of answers. Experiments in two benchmarks demonstrated the effectiveness of the proposed multimodal COT.

**Strengths:**

1.	This paper proposed a multimodal CoT reasoning framework by fusing the vision features extracted by ViT with the language features, which can mitigate the challenge of hallucination.
2.	This paper separated the CoT reasoning process into two stages: rationale generation and answer inference.
3.	This paper conducted extensive experiments and analysis. Experiment results demonstrated the effectiveness of the proposed methods.

**Weaknesses:**

1.	The performance of Multimodal-CoT falls behind some baselines, e.g. LLaVa (https://arxiv.org/abs/2304.08485) on the ScienceQA dataset and LXMERT on the AOKVQA dataset (https://aclanthology.org/D19-1514.pdf).
2.	It would be better to add more explanation or motivation about separating the reasoning process into two-stage works.

**Questions:**

Typos Grammar Style And Presentation Improvements:
1.	In section 4.2, “frozon” in “we fetch the patch-level features by frozon vision” should be amended to “frozen”.
2.	The Multimodal-CoT accuracy demonstrated in Table 9 and Table 10 is different from it in Table 4. Why?

---

> ### Author Response · Authors · 2023-11-19
>
> Thanks for your insightful review and constructive feedback.
>
> > W1: “The performance of Multimodal-CoT falls behind some baselines.”
>
> It is reasonable that Multimodal-CoT slightly falls behind LLaVA as it is with 1/10 parameters compared with LLaVA. It is worth noting that Multimodal-CoT keeps state-of-the-art performance for over several months after release and LLaVA takes our approach as the major baseline for comparison.
>
> > W2: "It would be better to add more explanation or motivation about separating the reasoning process into two-stage works."
>
> Yes. We have added more explanation when introducing the model. The key motivation is the anticipation that the answer inference can leverage better generated rationales that are based on multimodal information. By separating the training stages, we are able to obtain more effective rationales by training a specific rationale generation model. Then, the effective rationales are used to predict the final answer. In contrast, as results reported in Table 2, modeling the reasoning process by combining the rationale and answer (RA or AR) would lead to inferior results.
>
> > Q: Typos Grammar Style And Presentation Improvements.
>
> Thanks for the comments. We have fixed the typo. We have updated the Appendix to comply with the latest version of the paper.

---

### Meta-Review · Area_Chair_pvdf · 2023-12-07

**Metareview:**

The paper presents an approach in the multimodal integration of language and vision modalities. Reviewers acknowledge that its strengths lie in the novel two-stage framework and the mitigation of hallucination issues. However, the paper focuses only on text and image modalities, neglecting other modalities like audio and video. Additionally, the evaluation is limited to only two benchmark datasets, which may not fully represent the range of reasoning tasks. Some claims, particularly regarding surpassing human performance, are considered overreaching and not sufficiently supported by rigorous experimentation. Reviewers note that the Multimodal-CoT's performance falls behind some baselines like LLaVa and LXMERT on specific datasets. The overall ratings from the reviewers place it at the borderline, with two borderline accepts and two borderline rejects. The AC checked all the related materials, and find the weakness identified by the reviewers are valid. Thus, the paper is rejected.

**Justification For Why Not Higher Score:**

The paper focuses only on text and image modalities, neglecting other modalities like audio and video. Additionally, the evaluation is limited to only two benchmark datasets, which may not fully represent the range of reasoning tasks. Some claims, particularly regarding surpassing human performance, are considered overreaching and not sufficiently supported by rigorous experimentation. Reviewers note that the Multimodal-CoT's performance falls behind some baselines like LLaVa and LXMERT on specific datasets. The overall ratings from the reviewers place it at the borderline, with two borderline accepts and two borderline rejects.

**Justification For Why Not Lower Score:**

N/A

---

### Decision · Program_Chairs · 2024-01-16

Reject